# Design and Synthesis In Silico Drug-like Prediction and Pharmacological Evaluation of Cyclopolymethylenic Homologous of LASSBio-1514

**DOI:** 10.3390/molecules26164828

**Published:** 2021-08-10

**Authors:** Lidia Moreira Lima, Tiago Fernandes da Silva, Carlos Eduardo da Silva Monteiro, Cristiane Aparecida-Silva, Walfrido Bispo Júnior, Aline Cavalcanti de Queiroz, Magna Suzana Alexandre-Moreira, Gisele Zapata-Sudo, Eliezer J. Barreiro

**Affiliations:** 1Laboratório de Avaliação e Síntese de Substâncias Bioativas (LASSBio^®^), Instituto Nacional de Ciência e Tecnologia de Fármacos e Medicamentos (INCT-INOFAR), Universidade Federal do Rio de Janeiro, CCS, Cidade Universitária, Rio de Janeiro 21941-971, RJ, Brazil; tiagosilfer@yahoo.com.br (T.F.d.S.); cristianesilva486@gmail.com (C.A.-S.); 2Programa de Pós-Graduação em Farmacologia e Química Medicinal, Instituto de Ciências Biomédicas, Universidade Federal do Rio de Janeiro, Rio de Janeiro 21941-902, RJ, Brazil; kadusilvamonteiro@hotmail.com (C.E.d.S.M.); gzsudo@gmail.com (G.Z.-S.); 3Programa de Pós-Graduação em Química, Instituto de Química, Universidade Federal do Rio de Janeiro, Rio de Janeiro 21941-909, RJ, Brazil; 4Laboratório de Farmacologia Cardiovascular, Universidade Federal do Rio de Janeiro, Rio de Janeiro 21941-971, RJ, Brazil; 5LaFI—Laboratório de Farmacologia e Imunidade, Instituto de Ciências Biológicas e da Saúde, Universidade Federal de Alagoas, Maceió 57072-900, AL, Brazil; walfridobispojr@hotmail.com (W.B.J.); aline.queiroz@arapiraca.ufal.br (A.C.d.Q.); suzana.magna@gmail.com (M.S.A.-M.)

**Keywords:** acylhydrazone, homologation, anti-inflammatory, analgesic, pain, in silico drug-like

## Abstract

Acylhydrazones are still an important framework to the design of new bioactive compounds. As treatment of chronic pain represents a clinical challenge, we decided to modify the structure of LASSBio-1514 (**1**), previously described as anti-inflammatory and analgesic prototype. Applying the homologation as a strategy for molecular modification, we designed a series of cyclopentyl- (**2a**–**e**), cyclobutyl- (**3a**–**e**), and cyclopropylacylhydrazones (**4a**–**e**) that were synthetized and evaluated in murine models of inflammation and pain. A comparison of their in silico physicochemical and drug-like profile was conducted, as well as their anti-inflammatory and analgesic effect. Compounds **4a** (LASSBio-1755) and **4e** (LASSBio-1757) displayed excellent in silico drug-like profiles and were identified as new analgesic lead-candidates in acute and chronic model of pain, through oral administration.

## 1. Introduction

Acylhydrazone (R-CONHN=CHR_1_) is a versatile framework described as an important privileged structure. Inspired in this framework, several bioactive compounds have been designed and synthetized by modifying the nature of the substituents linked to the acyl (R-CO-) and/or imine (-N=CHR_1_) subunits [1,2,3,4,5,6,7]. Conformational and stereochemistry features of this framework have been studied, suggesting the *s*-trans amide conformer and the *E* geometrical isomers as the most prevalent species [8,9,10]. Recently, Thota and coworkers discussed the role of acylhydrazones as drugs and drug-candidates [11]. Owing to the fact that chronic pain is still an unmet medical need, the analgesic effect is one of the most recurrent activities ascribed to acylhydrazones [12,13].

Previously, our group has described the synthesis and pharmacological evaluation of cyclohexyl-*N*-acylhydrazones as orally active, analgesic, and anti-inflammatory lead-candidates. Among them, a compound bearing the 4-pyridinyl subunit (LASSBio-1514), linked to imine moiety, displayed important antihyperalgesic activity in a Spinal Nerve Ligation (SNL) model [14].

In an attempt to study the impact of homologation strategy [15] on the biological properties and drug-like profile of the cyclohexyl-*N*-acylhydrazones, we describe here the design, synthesis, pharmacological evaluation, and in silico drug-like prediction of cyclopolymethylenic homologous of compound LASSBio-1514 (**1**).

The design conception of the new cyclopolymethylene homologues series was proposed by decreasing size of the cyclohexyl ring of the prototype LASSBio-1514 (**1**), by the elimination of one, two, and three methylenes to generate the cyclopentyl (**2a**), cyclobutyl (**3a**), and cyclopropyl-*N*-acylhydrazone (**4a**), respectively. Further, isosteric ring replacement [16,17] was performed, allowing the substitution of the 4-pyridinyl moiety (**a**) by a phenyl (**b**), 2-thienyl (**c**), ferrocenyl (**d**), and 4-dimethylaminophenyl (**e**) ring (Figure 1).

## 2. Results and Discussion

### 2.1. Chemistry

The target compounds were synthetized in two linear steps, exploring the cycloalkyl esters (**5**–**7**) as starting materials. The synthesis was based on the hydrazinolysis of the esters **5**–**7**, followed by the condensation of the hydrazide intermediates (**8**–**10**) with aromatic and heteroaromatic aldehydes, using the methodology previously described by da Silva and coworkers, to obtain the desired cycloalkyl-acylhydrazones (**2a**–**e**, **3a**–**e**, and **4a**–**e**) in good yields [14] (Scheme 1).

Regarding the stereochemistry of imine double bond, all compounds (**2a**–**e**, **3a**–**e**, and **4a**–**e**) were obtained as *E*-isomers (Appendix A). The duplication of the amide (CONH) and imine (N=CH) signals in the hydrogen nuclear magnetic resonance (^1^H-NMR) spectra suggested the presence of two rotamers in solution. Variable temperature ^1^H-NMR studies (25 °C, 60 °C, and 90 °C) were performed. At 90 °C, the two signals coalesced to a single peak, confirming the supposition of amide conformational isomers (*s*-cis and *s*-trans) (Appendix A).

As previously synthesized and identified by Bastos and coworkers [18], compound **4a** was assigned as the isomer *E* by X-ray powder diffraction studies. Unlike the mixture of amide conformers (*s*-cis and *s*-trans) identified in solution, the *s*-cis amide conformation was found by X-ray powder diffraction experiment [18] (Figure 2).

The comparative physicochemical (Table 1) and drug-like (Table 2) profile of LASSBio-1514 (1) and their inferior homologous (**2a**–**e**, **3a**–**e**, and **4a**–**e**) was predicted using Percepta—a commercial Software of ACD/Labs [19]. With the exception of compounds **2d**, **3d**, and **4d** which, due to their organometallic nature, cannot be predicted by Percepta, all other compounds comply with Lipinski’s rules [20] and Veber’ guidelines [21]. As depicted in Table 1, compounds were predicted as water soluble, except for compound **2b** (solubility = 0.01 mg/mL). Similar to the prototype **1**, the homologous series were foreseen as having an ideal partition coefficient, with logP values ranging from 1 to 3, with the exception of compound **4a**, which was predicted to have the lowest lipophilicity. Based on the literature data, high lipophilicity is also expected for compounds **2d**, **3d**, and **4d** compared to their isosteres **2b**, **3b**, and **4b**, as ferrocene (Fc) is more lipophilic than benzene (Bz) (LogP_Fc_ = 3.54 compared to LogP_Bz_ = 2,13; π_Fc_= 2.46 compared to π_Bz_ = 1.96) [22,23]. Considering the log of the acid dissociation constant (pKa) calculated for the compounds containing the 4-pyridine (**a**) and 4-dimethylaminophenyl (**e**) moieties, no significant change was found for the distribution coefficient (LogD) of the target compounds at pH 4.6 (duodenum) and pH 7.4 (blood).

The in-silico drug-like properties of **2a**–**c**, **3a**–**c**, **4a**–**c**, and **2**–**4e** revealed that all compounds have highly permeable profiles in Caco-2 cells, and were predicted as able to penetrate the central nervous system. They were foreseen to have moderate to high plasma protein binding ability (PPB ranging from 57% to 87%) and volume of distribution between 1.3 to 2 L. Given the good in silico aqueous solubility and good permeability, the compounds were projected to have high oral bioavailability (80–99%). Unfortunately, the software was not able to determine the metabolic stability (HLM), mutagenic (AMES), and cardiotoxic potential of the target compounds. The score values obtained fell into the method’s undefined range (0.33–0.67).

### 2.2. Pharmacological Activities

Acetic acid- and formalin-induced pain protocols were used in order to investigate the peripheral analgesic profile of the cycloalkyl-acylhydrazones (**2a**–**e**, **3a**–**e**, and **4a**–**e**) [24,25]. Table 3 shows the analgesic effect observed after oral administration of the target compounds, indomethacin, dipyrone, and LASSBio-1514 (100 µmol/kg). Cyclobutyl (**3a**) and cyclopropyl (**4a**) analogues demonstrated similar inhibitory effect on acetic acid-induced pain, or slightly better than the prototype LASSBio-1514. Inhibition of abdominal writhing induced by acetic acid of 75.4 ± 8.1, 94.3 ± 2.0, 79.4 ± 3.3, 70.8 ± 1.9, 77.8 ± 7.6, and 72.3 ± 0.9% for the homologous **2e**, **3b**, **3c**, **4b**, **4c**, and **4d**, respectively, confirmed the intense analgesic activity (Table 3). To reinforce the analgesic profile of **2a**–**e**, **3a**–**e**, and **4a**–**e**, they were evaluated in formalin-induced pain test in mice. Unlike the cyclohexyl (**1**), cyclopentyl (**2a**–**d**) and cyclopropylacylhydrazones (**4a**, **4d**–**e**), the cyclobutyl derivatives (**3a**–**e**) were unable to display activity at neurogenic phase of formalin test. However, in the early phase of formalin test, compounds **4a** and **4e** inhibited a hypernociceptive response greater than 60%, indicating a possible central analgesic activity. The assumption was not confirmed by hot plate test (data not shown). Cyclopentyl homologous **2a**–**e** were the most active in the late phase of the formalin test, indicating their great effect in inflammatory pain (Table 3).

In order to investigate the anti-inflammatory profile of compound **1** and their inferior homologous (100 μmol/kg, p.o), they were evaluated at carrageenan-induced peritonitis model [26]. In this assay, all compounds inhibited leukocyte infiltration, revealing their anti-inflammatory activity (Table 4)

Considering the great analgesic profile of compounds **4a** (LASSBio-1755) and **4e** (LASSBio-1757) and their anti-inflammatory activity, associated to their in silico drug-like profile, they were selected for further investigation. As shown in Figure 3, both compounds showed a reduction in pain induced by the formalin in a dose-dependent manner. In the inflammatory phase, LASSBio-1755 (**4a**) reduced the reactivity to 100.1 ± 7.4 s and 19.4 ± 5.3 s after oral administration of 30 and 100 umol/kg, respectively. The compounds were more effective than acetylsalicylic acid (ASA) in reducing the hyperalgesic response during the neurogenic phase of formalin test. Figure 4 shows the effect of LASSBio-1755 (**4a**) and LASSBio-1757 (**4e**) on the hyperalgesic response induced by carrageenan. Both compounds reduced the hyperalgesic response at a dose eight times lower than ASA (Figure 4).

Further, it was evaluated the antinociceptive effect of the cyclopropyl-*N*-acylhydrazones **4a** and **4e** in a chronic pain model (Figure 5). The compounds reduced the thermal hyperalgesia and mechanical allodynia induced by the spinal nerve ligation (SNL) in rats, recovering the withdrawal latency from 7.2 ± 0.3 to 11.2 ± 0.3 s, and threshold from 24.0 ± 1.1 to 38.0 ± 0.5 g after 14 days of oral administration of **4a** (100 umol/kg). Similar results were observed for **4e** (100 umol/kg, p.o.).

## 3. Experimental Section

### 3.1. Chemistry

#### 3.1.1. General Methods

NMR spectra were determined in deuterated chloroform or dimethyl sulfoxide containing ca. 1% tetramethylsilane as an internal standard, using a 200/50 MHz Bruker DPX-200, 300/75 MHz Varian Unity-300, and 400 MHz Varian—MR spectrometer. The progress of all reactions was monitored by thin layer chromatography, which was performed on 2.0 cm × 6.0 cm aluminum sheets pre-coated with silica gel 60 (HF-254, Merck, Darmstadt, Germany) to a thickness of 0.25 mm. The developed chromatograms were viewed under ultraviolet light at 254 nm. Merck silica gel (230–400 mesh) was used for column chromatography. Elemental analyses were carried out on a Thermo Scientific Flash EA 1112 Series CHN-Analyzer. Melting points were determined with a Quimis Q340.23 apparatus and are uncorrected. All described products showed 1H and 13C NMR spectra according to the assigned structures. All organic solutions were dried over anhydrous sodium sulfate, and all organic solvents were removed under reduced pressure in rotatory evaporator. HPLC for purity determinations were conducted using Shimadzu LC-20AD with a Kromasil 100-5C18 (4.6 mm × 250 mm) and a Shimadzu SPD-M20A detector at 254 nm wavelength. The solvent system for HPLC purity analyses was 70:30 acetonitrile:phosphate buffer solution at pH 7. The isocratic HPLC mode was used, and the flow rate was 1.0 mL/min.

#### 3.1.2. General Procedure for Preparation of Cycloalkyl-hydrazides **8**–**10**

Hydrazine hydrate (80%, 4 equivalents) was added to a solution of methyl cycloalkyl esters 5–7 (1.00 g) in absolute ethanol (4 mL). The reaction mixture was kept under reflux for 48 h. Then, the solvent was evaporated under reduced pressure and ice was added to the residue, resulting in a precipitate formation. The precipitate was filtered to give compounds 8–10 in good yields.

##### Cyclopentanecarbohydrazide (**8**)

Compound 8 was obtained in 81% yield as a white powder. Its melting point is 111–113 °C. The data for this compound are in agreement with previous reports (117–118 °C) [27].

##### Cyclobutanecarbohydrazide (**9**)

Compound 9 was obtained in 63% yield as a white powder. Its melting point is 79–81 °C. The data for this compound are in agreement with previous reports (79–80 °C) [28].

##### Cyclopropanecarbohydrazide (**10**)

Compound 10 was obtained in 59% yield as a white powder. Its melting point is 74–76 °C. The data for this compound are in agreement with previous reports (98–99 °C) [29].

General Procedure for Preparation of cycloalkyllacylhydrazones **2a**–**e**, **3a**–**e**, and **4a**–**e**

The corresponding aromatic or heteroaromatic aldehyde (2 mmol) was added to a solution of cycloalkylcarbohydrazides (**8**–**10**, 2 mmol) in absolute ethanol (10 mL). The mixture was stirred for 2h at room temperature. At the end of the reaction, the volume of ethanol was partially concentrated at reduced pressure, and the resulting mixture was poured into cold water. The precipitate was filtered out, dried under vacuum, and then the solid was washed with n-hexane and/or recrystallized from ethanol to give the target cycloalkyl-acylhydrazones (**2a**–**e**, **3a**–**e** and **4a**–**e**).

*(E)-N’-(pyridin-4-ylmethylene)cyclopentanecarbohydrazide* (LASSBio-1521, **2a**)

Compound **2a** was obtained as a white powder in 59% yield by condensation of cyclopentane-carbohydrazide (**8**) with isonicotinaldehyde, m.p. 139–140 °C.

^1^H-NMR (200 MHz) DMSO-d*_6_* (ppm): 11.60/11.46 (s,1H, CONH-), 8,62 (d, 1H, *J* = 4 Hz, H8/H12) 8.18/7.95 (s, 1H, N=CH), 7,61 (d, 1H, *J* = 4 Hz, H9/H11), 3.51/2.50 (m, 1H, H1), 1.61 (m, 8H, cycloalkyl).^13^C-NMR (50 MHz) DMSO-d*_6_* (ppm): 178,07/172.87 (NC=O), 150.75 (2C, C9/C11), 143.92/140.30 (1C, N=CH), 142.23 (1C, C7), 121.41 (2C, C8/C12), 43.72 (1C, C1), 30.47/29.83 (2C, C-2/C5), 26.24 (2C, C3/C4). IR (KBr) νmax (cm^−1^): 3179 (NH); 2952 (CH3); 1669 (C=O); 1595 (C=N); 1406 (Pyridine). % purity = 96.55% by HPLC-C18. (acetonitrile/water 7:3) (Rt = 4.75 min).

*(E)-N’-benzylidenecyclopentanecarbohydrazide* (LASSBio-1519, **2b**)

Compound **2b** was obtained as a white powder in 80% yield by condensation of cyclopentane-carbohydrazide (**8**) with benzaldehyde, m.p. 160–161 °C. The data for this compound are in agreement with previous reports (158–160 °C) [5].

^1^H-NMR (200 MHz) DMSO-d*_6_* (ppm): 11.35/11.16 (s, 1H, CONH-), 8.18/7.98 (s, 1H, N=CH), 7.64 (m, 2H, H8 e H12), 7.41 (m, 3H, H9/H10/H11), 3.49/2.64 (m, 1H), 1.64 (m, 8H). ^13^C-NMR (50 MHz) DMSO-d*_6_* (ppm): 177.73/172.54 (NC=O), 146/142 (N=CH), 134,01 (1C, C7), 130,33 (2C, C8/C12), 127.03 (3C, C9/C10/C12), 43.43 (1C, C1), 29–31 (2C, C-2/C5), 26,25 (2C, C3/C4). IR [KBr] νmax (cm^−1^): 3187 (NH); 2953 (CH_3_); 1649 (C=O); 1562 (N=C). Anal. Calcd. for C13H16N2O: C, 72.19; H, 7.46; N, 12.95. Found: C, 72.11; H, 7.44; N 12.75.

*(E)-N’-(thiophen-2-ylmethylene)cyclopentanecarbohydrazide* (LASSBio-1520, **2c**)

Compound **2c** was obtained as a white powder in 71% yield by condensation of cyclopentane-carbohydrazide (**8**) with thiophene-2-carbaldehyde, m.p. 182–184 °C.

^1^H-NMR (200 MHz) DMSO-d*_6_* (ppm): 11.30/11.13 (s,1H, CONH-), 8.39/8.15 (s, 1H, N=CH), 7,63 (d, 1H, *J* = 5 Hz, H8), 7,46 (d, 1H, *J* = 5 Hz, H10), 7,34 (m, 1H, H9), 2,61 (m, 1H), 1.61 (m, 8H, cycloalkyl). ^13^C (50 MHz) DMSO-d*_6_* (ppm): 176.99/171.98 (NC=O), 141.22 (1C, C7), 139.38 (1C, C8), 137.49 (1C, C10), 130.53 (1C, C9), 127.95 (N=CH), 43.24 (1C, C1), 30.06/29.30 (2C, C-2/C5), 25.78 (2C, C3/C4).IR [KBr] νmax (cm^−1^): 3180(NH); 2948(CH3); 1653(C=O). Anal. Calcd. for C11H14N2O: C, 59.43; H, 6.35;N, 12.44. Found: C, 59.20; H, 6.30; N 12.44.

*(E)-N’-**(E)-N’-Ferrocenylcyclopentanecarbohydrazide* (LASSBio-1522, **2d**)

Compound **2d** was obtained as a white powder in 69% yield by condensation of cyclopentane-carbohydrazide (**8**) with ferrocenecarboxaldehyde m.p. 204–205 °C.

^1^H- NMR (200 MHz) DMSO-d*_6_* (ppm): 11.02/10.83 (s,1H, CONH-), 8.00/7.79 (s, 1H, N=CH), 4.59 (s, 2H, H7/H11), 4.40 (s, 2H, H9/H10), 4.20 (s, 5H), 3.51/2.50 (m, 1H, H1), 1.61 (m, 8H). ^13^C (50 MHz) DMSO-d*_6_* (ppm): 176.94/169.81 (NC=O), 148.24/143.28 (1C, N=CH), 79.84 (1C, C7), 70.72 (2C, C8/C11), 69.24 (2C, C9/C10), 68.07 (5C), 43.91 (1C, C1), 30.56/29.83 (2C, C-2/C5), 26.70 (2C, C3/C4). IR [KBr] νmax (cm^−1^): 3181 (NH); 2953 (CH3); 1649 (C=O); 1578 (C=N). Anal. Calcd. for C17H20FeN2O: C, 62.; 98H, 6.22;N, 8.64. Found: C, 61.59; H, 6.12; N 8.60.

*(E)-N’-(4-(dimethylamino)benzylidene)cyclopentanecarbohydrazide* (LASSBio-1518, **2e**)

Compound **2e** was obtained as a white powder in 71% yield by condensation of cyclopentane-carbohydrazide (**8**) with 4-(dimethylamino)benzaldehyde, m.p. 178–180 °C.

^1^H-NMR (200 MHz) DMSO-d*_6_* (ppm): 11/11.86 (s,1H, CONH-), 8.02/7.84 (s, 1H, N=CH), 7,49 (d, 2H, *J* =8 Hz, H11/H9), 6.75 (d, 2H, J= 8Hz, H8/H12), 3.34/2.51 (m, 1H), 2.96 (s, 6H), 1.71 (m, 8H, cycloalkyl). ^13^C-NMR (50 MHz) CDCl*_3_* (ppm): 160 (NC=O), 147 (N=CH), 128.73 (2C, C11/C9), 122.33 (2C, C7 e C10), 112.45 (2C, C8/C12), 41.34 (1C, C1), 30.57/29.81 (2C, C2/C5), 26,25 (2C, C3/C4). IR (KBr) νmax (cm^−1^): 3201 (NH); 2926 (CH_3_); 1663 (C=O); 1600 (N=C). Anal. Calcd. for C15H21FeN3O: C, 64.90; H, 8.43; N, 15.15. Found: C, 64.88; H, 8.43; N 14.93.

*(E)-N’-(pyridin-4-ylmethylene)cyclobutanecarbohydrazide* (LASSBio-1686, **3a**)

Compound **3a** was obtained as a white powder in 22% yield by condensation of cyclobutanecarbohydrazide (**9**) with isonicotinaldehyde, m.p. 121–123 °C.

^1^H-NMR (200 MHz) DMSO-d*_6_* (ppm): 11.44/11.36 (s, 1H, CONH-), 8.52 (d, 2H, *J* = 4 Hz, H8/H9), 8.08/7.83 (s, 1H, N=CH), 7.51 (d, 2H, *J* = 4 Hz, H7 e H10), 3.71/3.05 (m, 1H, H1), 2.15–1.65 (m, 6H). ^13^C-NMR (50 MHz ) DMSO-d*_6_* (ppm):176.58/171.38 (NC=O), 150.74 (2C, C8 e C9), 144.11/140.38 (s, N=CH), 142.12 (1C, C6), 121.41 (2C, C7 e C10), 36.66 (1C, C1), 24.95 (2C, C2 e C4), 18.48 (1C, C3). IR [KBr] νmax (cm^−1^): 3185 (NH); 2936 (CH_3_); 1671 (C=O); 1593 (C=N); 1406/807 (4-Pyridine). % purity = 98.97% by HPLC-C18. (acetonitrile/water 7:3) (Rt = 2.97 min)

*(E)-N’-benzylidenecyclobutanecarbohydrazide* (LASSBio-1687, **3b**)

Compound **3b** was obtained as a white powder in 62% yield by condensation of cyclobutanecarbohydrazide (**9**) with benzaldehyde, m.p. 124–126 °C.

^1^H (200 MHz) DMSO-d*_6_* (ppm): 11.20/11.14 (s, 1H, CONH-), 8.16/7.93 (s, 1H, N=CH), 7.64 (m, 2H, H7 e H11), 7.42 (m, 3H, H8/H9/H10), 4.40–3.11 (m, 1H, H1), 2.17–1.65 (m, 6H). ^13^C-NMR (50 MHz) DMSO-d*_6_* (ppm): 176 (NC=O), 142.22 (s, N=CH), 134.11 (1C, C6), 129.47 (2C, C7/C11), 126.50 (3C, C8/C9/C10), 37.08 (1C, C1), 24.36 (2C, C2/C4), 17.91(2C, C3). IR [KBr] νmax (cm^−1^): 3185 (NH); 2929 (CH_3_); 1651(C=O); 758/695 (phenyl). Anal. Calcd. for C12H14N2O: C, 7126.; H, 6.98; N, 13.85. Found: C, 71.00; H, 6.93; N 13.57.

*(E)-N’-(thiophen-2-ylmethylene)cyclobutanecarbohydrazide* (LASSBio-1685, **3c**)

Compound **3c** was obtained as a white powder in 55% yield by condensation of cyclobutanecarbohydrazide (**9**) with thiophene-2-carbaldehyde, m.p. 128–130 °C.

^1^H-NMR (200 MHz) DMSO-d*_6_* (ppm): 11.15/11.13 (s, 1H, CONH-), 8.40/8.10 (s, 1H, N=CH), 7.63 (d, 2H, *J* = 5 Hz, H9), 7.39 (d, 2H, *J* = 5Hz, H7), 7.13 (d, 1H, *J* = 5Hz, H9), 3.81/3.05 (m, 1H, H1), 2.15–1.65 (m, 6H, cycloalkyl). ^13^C-NMR (50 MHz) DMSO-d*_6_* (ppm): 176.58/171.38 (NC=O), 150.74 (2C, C7), 144.11/140.38 (s, N=CH), 142.12 (1C, C6), 121.41 (2C, C8 e C9), 36.66 (1C, C1), 24.95 (2C, C2 e C4), 18.48 (1C, C3). IR [KBr] νmax (cm^−1^): 3167 (NH); 2949 (CH3); 1663 (C=O); 1593 (C=N); 1441 (2-thiophene). % purity = 99.41% by HPLC-C18. (acetonitrile/water 6:4) (Rt = 9.74 min).

*(E)-N’-Ferrocenylcyclobutanecarbohydrazide* (LASSBio-1683, **3d**)

Compound **3d** was obtained as a white powder in 60% yield by condensation of cyclobutanecarbohydrazide (**9**) with ferrocenecarboxaldehyde, m.p. 92–94 °C.

^1^H-NMR (200 MHz) DMSO-d*_6_* (ppm): 10.87/10.82 (s,1H, CONH-), 8.00/7.74 (s, 1H, N=CH), 4.58 (s, 2H, H6/H9), 4.40 (s, 2H, H7/H8), 141 4.20 (s, 5H, H10), 3.67/3.02 (m, 1H, H1), 2.15–1.65 (m, 6H). ^13^C-NMR (50 MHz) DMSO-d*_6_* (ppm): 174.92/169.54 (NC=O), 147.31/142.73 (s, N=CH), 69.91 (2C, C6/C9), 68.81 (2C, C7/C8), 67.08 (5C, C10), 36.11 (1C, C1), 24.41 (2C, C2 e C4), 17.92 (1C, C3). % purity = 99.17% by HPLC-C18. (acetonitrile/water 7:3) (Rt = 5.19 min).

IR [KBr] νmax (cm^−1^): 3440 (NH); 3130 (CH3); 1660 (C=O). % purity = 99.17% by HPLC-C18(acetonitrile/water 7:3) (Rt = 5.19 min)

*(E)-N’-(4-(dimethylamino)benzylidene)cyclobutanecarbohydrazide* (LASSBio-1684, **3e**)

Compound **3e** was obtained as a white powder in 50% yield by condensation of cyclobutanecarbohydrazide (**9**) with 4-(dimethylamino)benzaldehyde, m.p. 186–188 °C.

^1^H (200 MHz) CDCl*_3_* (ppm): 9.75/8.95 (s, 1H, CONH-), 8.04/8.10 (s, 1H, N=CH), 7.59 (d, 2H, *J* = 8 Hz, H7/H10), 6.72 (d, 2H, *J* = 8 Hz, H7 e H11), 3.01 (s, 6H, CH3), 3.81/3.05 (m, 1H, H1), 2.15–1.65 (m, 6H). ^13^C-NMR (50 MHz) CDCl_3_ (ppm): 177.35 (NC=O), 151.56 (1C, C8/C9), 143.99 (s, N=CH), 129.22 (2C, C7/C11), 122.10 (2C, C7), 111.75 (2C, C8/C10), 40.29 (2C, CH3), 36.88 (1C, C1), 25.06 (2C, C2 e C4), 18.48 (1C, C3). IR [KBr] νmax (cm^−1^): 3200 (NH); 2937 (CH3); 1665 (C=O); 1599 (C=N); 1367 (ArN(CH3)2). Anal. Calcd. for C14H19N3O: C, 63.84; H,7.23; N, 15.96. Found: C, 64.02; H, 7.88; N 15.82.

*(E)-N’-(pyridin-4-ylmethylene)cyclopropanecarbohydrazide* (LASSBio-1755, **4a**)

Compound **4a** was obtained as a white powder in 42% yield by condensation of cyclopropanecarbohydrazide (**10**) with isonicotinaldehyde, m.p. 193–194 °C, as previously described by Bastos and coworkers (2017) [18].

^1^H-NMR (200 MHz) DMSO-d*_6_* (ppm): 8.60 (d, 2H, *J* = 4 Hz, H7/H8), 8.16/8.00 (s, 1H, N=CH), 7.60 (d, 2H, *J* = 4 Hz, H6/H9), 2.67/1.64 (m, 1H, H1), 0.99–0.81 (m, 4H). ^13^C-NMR (50 MHz) DMSO-d*_6_* (ppm): 175.00/169.83 (NC=O), 159.63/142.98 (s, N=CH), 150.31/150.16 (1C, C7/C8), 142.98/141.65 (1C, C5), 122.81 (2C, C6/C9), 126.89 (2C, C7/C9), 12.94/9.72 (1C, C1), 8.12/7.25 (2C, C2/C3). IR [KBr] νmax (cm^−1^): 3176 (NH); 2952 (CH_3_); 1663(C=O); 1595 (C=N). % purity = 99.95% by HPLC-C18. (acetonitrile/water 7:3) (Rt = 2.23 min).

*(E)-N’-benzylidenecyclopropanecarbohydrazide* (LASSBio-1753, **4b**)

Compound **4b** was obtained as a white powder in 55% yield by condensation of cyclopropanecarbohydrazide (**10**) with benzaldehyde, m.p. 152–154 °C. The data for this compound are in agreement with previous reports (152 °C) [6].

^1^H-NMR (200 MHz) DMSO-d*_6_* (ppm): 11.63/11.37 (s, 1H, CONH-), 8.18/8.04 (s, 1H, N=CH), 7.70 (m, 2H, H6/H10), 7.42 (m, 3H, H7/H8/H9), 2.73/1.67 (m, 1H, H1), 1.09–0.65 (m, 4H). ^13^C-NMR (50 MHz) DMSO-d*_6_* (ppm): 174.57/169.32 (NC=O), 145.36/142.77 (s, N=CH), 134.32 (1C, C5), 129.76 (1C, C8), 128.73 (2C, C6/C10), 126.89 (2C, C7/C9), 12.83/9.70 (1C, C1), 7.81/6.89 (2C, C2/C3). IR [KBr] νmax (cm^−1^): 3189 (NH); 3013 (CH3); 1645 (C=O); 757/692 (phenyl). % purity = 99.80% by HPLC-C18. (acetonitrile/water 7:3) (Rt = 3.73 min).

*(E)-N’-(thiophen-2-ylmethylene)cyclopropanecarbohydrazide* (LASSBio-1756, **4c**)

Compound **4c** was obtained as a white powder in 53% yield by condensation of cyclopropanecarbohydrazide (**10**) with thiophene-2-carbaldehyde, m.p. 137–140 °C.

^1^H-NMR (200 MHz) DMSO-d*_6_* (ppm): 11.57/11.35 (s, 1H, CONH-), 8.40/8.22 (s, 1H, N=CH), 7.63 (d, 1H, *J* = 5 Hz, H8), 7.43 (d, 1H, *J* = 5 Hz, 144 H6), 7.14 (dd, 1H, *J* = 5 Hz, *J* = 2 Hz, H7), 2.51/1.65 (m, 1H, H1), 0.99–0.77 (m, 4H). ^13^C-NMR (50 MHz) DMSO-d*_6_* (ppm): 174.23/169.18 (NC=O), 140.66/138.01 (s, N=CH), 139.19/139.06 (1C, C5), 130.49/129.94 (1C, C8), 128.48/127.99 (1C, C6), 127.84/127.74 (1C, C7), 12.86/9.53 (1C, C1), 7.86/6.94 (2C, C2/C3). IR [KBr] νmax (cm^−1^): 3188 (NH); 2939 (CH3); 1660 (C=O). % purity = 98.45% by HPLC-C18. (acetonitrile/water 7:3) (Rt = 3.53 min).

*(E)-N’-Ferrocenylcyclopropanecarbohydrazide* (LASSBio-1754, **4d**)

Compound **4d** was obtained as a white powder in 56% yield by condensation of cyclopropanecarbohydrazide (**10**) with ferrocenecarboxaldehyde, m.p. 223–225 °C. The data for this compound are not in agreement with previous reports (208–209 °C) [6].

^1^H-NMR (200 MHz) DMSO-d*_6_* (ppm): 11.30/11.05 (s, 1H, CONH-), 8.01/7.85 (s, 1H, N=CH), 4.69 (s, 2H, H5/H8), 4.31 (s, 2H, H6/H7), 4.07 (s, 5H, H9), 2.73/1.57 (m, 1H, H1), 1.09–0.65 (m, 4H). ^13^C-NMR (50 MHz) DMSO-d*_6_* (ppm): 69.93 (2C, C6/C7), 68.86(2C, C5/C8), 66.08 (5C, C9), 12.03/9.27 (1C, C1), 7.81/6.89 (2C, C2/C3). IR [KBr] νmax (cm^−1^): 3181 (NH); 3005 (CH3); 1653 (C=O). % = 95.39% by HPLC-C18.(acetonitrile/water 7:3) (Rt = 4.37 min).

*(E)-N’-(4-(dimethylamino)benzylidene)cyclopropanecarbohydrazide* (LASSBio-1757, **4e**)

Compound **4e** was obtained as a white powder in 47% yield by condensation of cyclobutanecarbohydrazide (**10**) with 4-(dimethylamino)benzaldehyde, m.p. 184–186 °C.

^1^H-NMR (200 MHz) DMSO-d*_6_* (ppm): 11.31/11.07 (s,1H, CONH-), 8.50/8.03 (s, 1H, N=CH), 7.67 (d, 2H, *J* = 8Hz, H6/H10), 6.74 (d, 2H, *J* = 8Hz, H8/H9), 3.38 (s, 6H, CH3) 2.67/1.63 (m, 1H, H1), 0.85–0.73 (m, 4H). ^13^C-NMR (50 MHz) DMSO-d*_6_* (ppm): 174.01/168.69 (NC=O), 151.35/151.20 (1C, C8), 146.20/143.66 (s, N=CH), 128.21/127.88 (2C, C6/C10), 121.76/121.70 (1C, C5), 111.81 (2C, C7/C9), 40.29 (2C, CH3), 12.78/9.69 (1C, C1), 7.59/6.59 (2C, C2/C3). IR [KBr] νmax (cm^−1^): 3190 (NH); 2924 (CH3); 1656 (C=O); 1611 (C=N). % purity = 95.87% by HPLC-C18. (acetonitrile/water 7:3) (Rt = 2,67 min).

### 3.2. Antinociceptive and Anti-Inflammatory Pharmacological Evaluation

#### 3.2.1. Animals

Male Swiss mice (20–30 g) were obtained from Federal University of Rio de Janeiro and BIOCEN-UFAL, and male Wistar rats (180–220 g) were obtained from Federal University of Rio de Janeiro. Animals was housed in group cages and maintained on a 12 h light/12 h dark cycle. The animals had free access to food and water at all times. Experiments were carried out according to a protocol approved by the Animal Welfare Committee of Federal University of Alagoas (UFAL) (Protocol Number: 026681/2009-23), and Institutional Animal Care and Use Committee at Universidade Federal of Rio de Janeiro (UFRJ) and in accordance with the ethical guidelines for investigation of experimental pain in conscious animals.

#### 3.2.2. Reagents

Acetic acid (Merck, Rio de Janeiro, Brasil), gum arabic (Sigma-Aldrich, Rio de Janeiro, RJ, Brasil), dipyrone (Sigma-Aldrich), and indomethacin (Sigma-Aldrich) were obtained from commercial sources. A solution of formalin 2.5% was prepared with formaldehyde (Merck) in saline (NaCl 0.9%). Dimethyl sulfoxide (DMSO, used as vehicle) and tramadol were donated by Cristália Produtos Químicos e Farmacêuticos Ltd.a (Itapira, SP, Brazil).

#### 3.2.3. Carrageenan-Induced Peritonitis

Peritoneal inflammation was induced according to the method described by Ferrandiz and Alcaraz [20]. A solution of carrageenan 1% (Sigma-Aldrich) was prepared in saline (NaCl 0.9%) and injected into the peritoneal cavity of mice (250 μL/animal). After 4h of carrageenan injection, the animals were euthanized by cervical dislocation and the peritoneal cavity was washed with 3 mL of cold Hank’s. Compounds and indomethacin were administered at the dose of 100 μmol/kg (p.o.) 30 min before carrageenan injection. The control group received 10 mL/kg of the vehicle (gum arabic, p.o.). The number of cells was quantified by optical microscope, using 100× lens.

#### 3.2.4. Acetic Acid-Induced Writhing Test

Mice received intraperitoneally (i.p.) administered acetic acid (0.6%, *v*/*v*, 0.1 mL/10 g), as previously reported [24,30]. The number of writhes, a response consisting of contraction of an abdominal wall, pelvic rotation followed by hind limb extension, was counted during continuous observation for 20 min beginning from 5 min after the acetic acid injection. Dipyrone and compounds (all 100 μmol/kg, oral administration) were administered 60 min before the acetic acid injection. Antinociceptive activity was expressed as inhibition percent of the usual amount of writhing observed in control animals.

#### 3.2.5. Formalin Induced Nociception

The procedure was performed as described by Sudo et al. (2015) [30] in mice to investigate the antinociceptive effects of LASSBio-1755 (4a) and LASSBio-1757 (4e) on neurogenic and inflammatory pain. The formalin (2.5%, 20 µL) was injected into the right hind paw 15 min after oral administration of DMSO (50 µL, vehicle), acetylsalicylic acid (150 mg/kg, reference drug), LASSBio-1755 (30 and 100 µmol/kg) and LASSBio-1757 (30 and 100 µmol/kg). Evaluation started immediately after formalin injection, being evaluated during 0–5 min (neurogenic pain response) and 15–30 min (inflammatory pain response). The duration of licking and biting of the injected paw was analyzed.

#### 3.2.6. Carrageenan-Induced Nociception

The thermal hyperalgesia induced by carrageenan (1%, 20 µL) was evaluated in mice according to the method described by Mendes et al. (2009) [31]. Peripheral inflammation was caused by intraplantar injection of carrageenan into the right hind paw in mice. The latency of each animal to the thermal stimuli was assessed at different times: before (control) and after carrageenan injection. Saline, acetylsalicylic acid (830 µmol/kg), and LASSBio-1755 (4a) (100 µmol/kg) and LASSBio-1757 (4e) (100 µmol/kg) were orally administered 15 min before carrageenan. Heat stimulus under light applied on hind paws until the paw withdrawal was studied; the time between thermal stimulus and paw withdrawal defined as latency. A cut-off time of 15 s used to avoid tissue damage. Evaluation was made using a plantar analgesia meter (ITC Inc. model 33).

#### 3.2.7. Neuropathic Pain Model Induced by Spinal Nerve Ligation

The peripheral neuropathy model was induced by spinal nerve ligation (SNL), according with Kim and Chung (1992) [26]. Wistar rats (200–220 g) were anesthetized with ketamine (100 mg/kg, i.p.) and xylazine (5 mg/kg, i.p.), and endured an incision of approximately 1 cm at the level of spinal L5 to S1. The L6 transverse process was partially removed to allow identification of L4 and L5 spinal nerves. The right L5 spinal nerve was isolated and ligated with 6.0 silk. The false-operated animals (SHAM) used for control were submitted to the same surgery procedure without the nerve ligation.

##### Evaluation of the Thermal Hyperalgesia and Mechanical Allodynia Induced by SNL

Thermal hyperalgesia and mechanical hypersensitivity signals were defined as a significant reduction in the paw withdrawal threshold and latency, respectively, in SNL animals compared to SHAM. After the development of these signals, the animals were orally treated for 7 days with LASSBio-1755 (100 µmol/kg), LASSBio-1757 (100 µmol/kg), tramadol (33 µmol/kg) or DMSO (100 µL, vehicle). For measurement of paw withdrawal latency [32], we used a radiant heat source (Ugo Basile model 37,370), applied the plantar surface of the hind paws, taking three measurements with a cut-off of 30 s to avoid tissue injury. However, the withdrawal threshold was evaluated using a digital version of Von Frey filaments (Analgesymeter Digital Device, model EFF301) [24,30]. Mechanical stimuli were applied to plantar region of the hind paw, and the withdrawal threshold was taken in five measurements with a 120 g cut-off to avoid tissue injury.

##### Statistical Analysis

Results are expressed as means ± standard error of means (SEM). The values of reactivity in the formalin test, paw withdrawal latency, and threshold were compared by Analysis of Variance One-Way followed by Newman–Keuls test. Two-way ANOVA, followed by Bonferroni post hoc was used to analyze carrageenan-induced paw withdrawal, using GraphPad Prism (version 5.0; GraphPad Software, Inc., San Diego, CA, USA). *p* values of <0.05 were considered significant.

## 4. Conclusions

Taken together, a systematic inferior homologation was applied at the structure of the previous prototype 1 (LASSBio-1514), yielding the design and synthesis of the cyclopentyl-, cyclobutyl- and cyclopropylacylhydrazones, which showed similar in silico physicochemical and drug-like profiles to the parent compound **1**. Compounds revealed a good in vivo anti-inflammatory and antinociceptive profile, allowing the identification of compounds 4e (LASSBio-1757) and 4a (LASSBio-1755, previously synthetized and characterized by Bastos and coworkers [18]), as new analgesic lead-candidates, active by oral administration in acute and chronic model of pain. Additionally, our data demonstrate the success of the homologation strategy in the design of bioactive compounds.

## Data Availability

Not applicable.

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
