# Peer review of "Design and Synthesis In Silico Drug-like Prediction and Pharmacological Evaluation of Cyclopolymethylenic Homologous of LASSBio-1514"

_molecules, 2021, doi:10.3390/molecules26164828_

Round 1
Reviewer 1 Report
This study is another endeavor to design new bioactive acylhydrazones. The use of chemoinformatics and insilico methods for the design is a good asset.
I recommend publishing the manuscript but after performing the following:
1- The title of the manuscript has unexplained spaces. Please adjust.
2- The English language of the whole manuscript requires a thorough revision.
3- The use of physico-chemical descriptors in predicting the chemical compounds behavior and other benefits has previously been utilized in:
Materials (Basel). 2018 Jul 1;11(7):1123. doi: 10.3390/ma11071123. Please refer to the use of these chemoinformatics methods and their benefits briefly in the introduction.
4- Please provide the full origin of the software ACD/Percepta and determine whether it is an open source or a commercial one.
Author Response
Reviewer 1
I recommend publishing the manuscript but after performing the following:
- The title of the manuscript has unexplained spaces. Please adjust.
Response: Done
- The English language of the whole manuscript requires a thorough revision.
Response: Done
3- The use of physico-chemical descriptors in predicting the chemical compounds behavior and other benefits has previously been utilized in:
Materials (Basel). 2018 Jul 1;11(7):1123. doi: 10.3390/ma11071123. Please refer to the use of these chemoinformatics methods and their benefits briefly in the introduction.
Response: We’ve read the paper: Shah, S., Firlak, M., Berrow, S., Halcovitch, N., Baldock, S., Yousafzai, B., … Hardy, J. (2018). Electrochemically Enhanced Drug Delivery Using Polypyrrole Films. Materials, 11(7), 1123. doi:10.3390/ma11071123. However, the description of the physical properties described in the article were calculated by a software (Molecular Operating Environment (MOE) software) different from the one used in the manuscript under analysis, and for compounds are equally different. For these reasons, authors understand that it would be inappropriate to quote this work in the article.
- Please provide the full origin of the software ACD/Percepta and determine whether it is an open source or a commercial one.
Response: Percepta is a commercial Software of ACD/Labs that Predict Molecular Properties/Physicochemical, ADME & Toxicity from Chemical Structure. This information was added as requested.
Reviewer 2 Report
Dear authors,
Good overall effort and nice results with potential that need to be necessarily enriched based on the bellow mentioned remarks:
Line 39: replace “de” with “the”
Line 41: replace “may due” with “maybe due to”
Line 49: omit –ed from described at the end of the sentence
Lines 52-57: Between these lines or in a new separate paragraph, since the authors mention design, it should be added the significance and rational distributed from the forrocene derivative. Because in terms of medicinal chemistry. Given the fact also that it was excluded from PK determination and no activity present why it was included? Maybe consider removing?
Line 69: Scheme 1 need to be revised because it presents an aziridine instead of a cyclopropyl ring with brackets on one methylene atom and an n indicator on the bottom right. Accordingly then n would be replaced with numbers n = 1-3 instead of -CH2-, -CH2CH2- and - CH2CH2CH2- which are now.
Lines 72-76: Very correctly rotamers are being mentioned by the authors but since experiments already have been performed maybe they could add an indicative example for just one compound at 45 & 90oC?
Lines 77-80: You should add along the paragraph “As previously synthesized and identified by Bastos et al, compound 4a [15] was assigned …” and rephrase accordingly thereafter.
Line 84: Please add a reference for ACD/Percepta.
Line 117: Replace “physico-chemistry” with “physicochemical”
Line 129: Replace “slightly superior” with “slightly better”
Line 135: Rephrase “exhibited an expressive action”
Lines 457 & 460: APA given format should be replaced accordingly…is this the reference 24 or another?
Line 472: Rephrase sentence…there is no word “obtention”
Additional remarks
- Compound 4a is being analyzed/studied herein like its being synthesized for the first time, which is not. It’s being used as a control? In continuation to reference 15? If yes it should be discussed accordingly.
- Figure S13 is missing the structure
- All supplementary material figures of NMR spectra should also include numbering of compound inside the figure under structure and also the temperature taken.
Author Response
Reviewer 2
Line 39: replace “de” with “the”
Response: Done
Line 41: replace “may due” with “maybe due to”
Response: Done
Line 49: omit –ed from described at the end of the sentence
Response: Done
Lines 52-57: Between these lines or in a new separate paragraph, since the authors mention design, it should be added the significance and rational distributed from the forrocene derivative. Because in terms of medicinal chemistry. Given the fact also that it was excluded from PK determination and no activity present why it was included? Maybe consider removing?
Response: Ferrocene (Fc) is a well-known isostere of phenyl ring, the two most successful examples of this king of isosteric replacement are ferrocifen and ferroquine (FQ, also known as SR97193). Patra, M., Gasser, G. The medicinal chemistry of ferrocene and its derivatives. Nat Rev Chem 1, 0066 (2017). https://doi.org/10.1038/s41570-017-0066.
The authors do not intend to remove the compounds containing ferrocene (2d, 3d and 4d). Although the software employed is not parameterized to calculate the properties of organometallic compounds, our in vivo data show that compounds 2d, 3c and 3d had very similar activity to the compound containing the phenyl ring (2b, 3b and 4b), as expected given the previous description of an isosteric relationship between benzene and ferrocene. We appreciate the suggestion and concern. We will include in the text two references about the use of ferrocene in medicinal chemistry.
Line 69: Scheme 1 need to be revised because it presents an aziridine instead of a cyclopropyl ring with brackets on one methylene atom and an n indicator on the bottom right. Accordingly then n would be replaced with numbers n = 1-3 instead of -CH2-, -CH2CH2- and - CH2CH2CH2- which are now.
Response: Done
Lines 72-76: Very correctly rotamers are being mentioned by the authors but since experiments already have been performed maybe they could add an indicative example for just one compound at 45 & 90oC?
Response: We have added Figure S31 which gives the 1H NMR spectra of compound 4c at 25°C, 60°C and 90°C.
Lines 77-80: You should add along the paragraph “As previously synthesized and identified by Bastos et al, compound 4a [15] was assigned …” and rephrase accordingly thereafter.
Response: Done
Line 84: Please add a reference for ACD/Percepta.
Response: There is no typical description reference for this software. But we added as a reference the link to Percepta's home page.
Line 117: Replace “physico-chemistry” with “physicochemical”
Response: Done
Line 129: Replace “slightly superior” with “slightly better”
Response: Done
Line 135: Rephrase “exhibited an expressive action”
Response: Done
Lines 457 & 460: APA given format should be replaced accordingly…is this the reference 24 or another?
Response: Done
Line 472: Rephrase sentence…there is no word “obtention”
Response: Done
Additional remarks
- Compound 4a is being analyzed/studied herein like its being synthesized for the first time, which is not. It’s being used as a control? In continuation to reference 15? If yes it should be discussed accordingly.
Response: Among the compounds in the series, only the 4a was revealed by our research group in the paper by Bastos et al. [15] in order to describe its characterization by X-ray diffraction. In this work, we describe for the first time the design of all the series and the biological data obtained with 4a and its other isosteres. To contemplate the referee's suggestion, we added the following sentence to the conclusions: “identification of compounds 4e (LASSBio-1757) and 4a (LASSBio-1755, previously synthetized and characterized by Bastos and coworkers [15])”
- Figure S13 is missing the structure
Response: Done
- All supplementary material figures of NMR spectra should also include numbering of compound inside the figure under structure and also the temperature taken.
- Response: Done
Reviewer 3 Report
In this manuscript Lima and coworkers describe the development of analgesic/anti-inflammatory acylhydrazones, analogues of a derivative previously described (LASSBio-1514). Several compounds show interesting biological activity in rodent models of pain and inflammation. Overall the results and findings are worthy of a publication in Molecules, although major revisions are required.
- The use of the English language is poor and needs significant revision, especially in the introduction section which is very difficult to follow. Other examples of poor English language are the paragraph between lines 77-80 (where is not clear if the X-ray experiment were performed by the authors or not) and Figure 1 (the words "among others" are wrong in this context). I strongly recommend a consult with a native english speaker or availing a professional scientific editing service before resubmitting the manuscript.
- Scheme 1: the aziridine ring depicted should be replaced with a carbocycle with a lower case n.
- Line 74: The NMR spectra of the variable temperature NMR experiment should be given in the supplementary material.
- Figure 2: the s-cis conformer should have a "Z" letter on the double bond.
- The solubility of the compounds should be measured experimentally. This is a simple experiment to perform and the values predicted by the software are generally scarcely reliable.
- Several compounds have already been described before (e.g. J. Het Chem. 2016, 53, 393-402; J. Chem. Eng. Data, 1963, 8,4,604: Int. Patent Application. WO1986004582). These reference should be added together with any other relevant one.
- A conculsion section summing up the results and the significance of the work should be added.
Once the above-mentioned revisions are made, I have no further reservations in recommending the publication of the work in Molecules.
Author Response
In this manuscript Lima and coworkers describe the development of analgesic/anti-inflammatory acylhydrazones, analogues of a derivative previously described (LASSBio-1514). Several compounds show interesting biological activity in rodent models of pain and inflammation. Overall the results and findings are worthy of a publication in Molecules, although major revisions are required.
- The use of the English language is poor and needs significant revision, especially in the introduction section which is very difficult to follow. Other examples of poor English language are the paragraph between lines 77-80 (where is not clear if the X-ray experiment were performed by the authors or not) and Figure 1 (the words "among others" are wrong in this context). I strongly recommend a consult with a native english speaker or availing a professional scientific editing service before resubmitting the manuscript.
Response: Done
- Scheme 1: the aziridine ring depicted should be replaced with a carbocycle with a lower case n.
Response: Done
- Line 74: The NMR spectra of the variable temperature NMR experiment should be given in the supplementary material.
Response: Done
- Figure 2: the s-cis conformer should have a "Z" letter on the double bond.
Response: The imine double bond geometry of the compound exemplified in Figure 2 is E. The electron pair of nitrogen is opposite to hydrogen, i.e., the two lighter substituents are on opposite sides, hence isomery E.
- The solubility of the compounds should be measured experimentally. This is a simple experiment to perform and the values predicted by the software are generally scarcely reliable.
Response: We totally agree with the referee, but unfortunately we are unable to carry out these experiments. Our laboratory has a reduced work routine due to restrictions imposed by COVID-19.
- Several compounds have already been described before (e.g. J. Het Chem. 2016, 53, 393-402; J. Chem. Eng. Data, 1963, 8,4,604: Int. Patent Application. WO1986004582). These references should be added together with any other relevant one.
Response: References J. Het Chem. 2016, 53, 393-402; J. Chem. Eng. Data, 1963, 8,4,604 were added at experimental section. Unfortunately, we were unable to find the compounds in this manuscript among the thousands claimed in the patent: WO1986004582 - ANTHELMINTIC PYRIDINYL ACYLHYDRAZONES, METHOD OF USE AND COMPOSITIONS.
- A conculsion section summing up the results and the significance of the work should be added.
Response: Done, thank you for the suggestion and corrections.
Round 2
Reviewer 2 Report
Dear authors,
the manuscript after all the implemented revisions is providing a more detailed view to the readers. Therefore it is ready for publication after a small correction to scheme 1(Line 70). For your assistance please mind the attached file named "Scheme 1_correction sample" and revise accordingly.
Best regards

Reviewer 3 Report
The authors have adequately addressed my concerns. I don't have any further comment and I recommend the publication of the manuscript in the present form